# TBOE maturity assessment model for standard digitalization: An empirical analysis using AHP and Delphi method

Xize Liu[1,¤a], Bingyan Zhang[1,¤a*], Yiyi Wang[1,¤a], Nana Niu[1,¤a], Jingsheng Li[1,¤a], Xinrui Hu[2,¤b]

**1** National Library of Standards (Research Center for Standards Digitalization)/China National Institute of Standardization, Beijing, China, **2** Software Product Quality Inspection Centre/Anhui Institute of Quality and Standardization, Hefei, China

¤a Current address: No. 4 Zhichun Road, Haidian District, Beijing, China
¤b Current address: No. 112 Ningguo Road, Baohe District, Hefei City, Anhui Province, China
* zhangbingy@cnis.ac.cn

## Abstract

With the advancement of digital, networked, and intelligent transformations in the economy and society, the integration of standardization with digital technology has garnered significant attention globally. Consequently, the digitization of standards has become an inevitable trend. This study aims to address the challenges in standard digitization by proposing a comprehensive framework. Distinct from the traditional TOE framework, this study explicitly introduces the Business dimension as an independent analytical factor, constructing the novel TBOE (Technology-Business-Organization-Environment) model that better captures the business-driven nature of standard digitalization. Through factor analysis, key influencing factors are identified. A maturity assessment model is developed, categorizing the digitization process into three stages. The model incorporates rigorous indicator normalization procedures and stage division thresholds derived from the Gartner Hype Cycle framework. Sensitivity analysis confirms the robustness of the model under weight perturbations and measurement uncertainties. An empirical analysis within the power industry validates the model's applicability, providing theoretical guidance and a practical toolkit for standard digitization across various sectors.

## Introduction

The digital economy is becoming a pivotal force in reorganizing global resources, reshaping the economic structure, and changing the global competitive landscape. As a cross-integration field of standardization and digital technology, the standard digitalization bears the critical responsibility of fostering standardization within the digital economy.International standardization organizations such as ISO, IEC, CEN, CENELEC, along with developed nations including the United Kingdom, the United States, and Germany, have incorporated standard digitalization into their strategic

**Data availability statement:** The data underlying the results presented in the study are available from Zenodo (URL:https://zenodo.org/records/18885793).

**Funding:** This work was supported by the Science and Technology Project of the State Administration for Market Regulation (2025MK212 to XL) and the Special Fundamental Research Fund for the Central Public Scientific Research Institutes (292024Y-11794 to XL).

**Competing interests:** The authors have declared that no competing interests exist.

agendas, leading research and application efforts in industry, construction, and social governance [1–8].

As one of the world's major economies, China places significant emphasis on standard digitalization. In October 2021, the CPC Central Committee and The State Council issued the "Outline for national standardized development", which explicitly stated in its development goals to "develop machine-readable and open-source standards, and promote the digital, networked, and intelligent transformation of standardization work" [9]. In February 2023, they further issued the "Overall layout plan for the construction of Digital China", proposing to "build a technical standard system, compile work guidelines for digital standards, and accelerate the formulation and revision of application standards for digital transformation across various industries and for cross-industrial integration and development" [10].

In recent years, China has actively carried out work related to the standard digitalization. At the organizational level, by the end of 2022, the National Standards Committee set up the National Standards Digitization Standardization Working Group (SAC/SWG 29), which is the secretariat of the China Institute of Standardization, responsible for the revision of national standards such as the basic generalization of standard digitalization, modeling, and realization of common technology and application technology. At the level of scientific research projects, in 2021 and 2022, the National Key RD Program "Common Technology and International Standards Research in Key Areas of Machine-readable Standards" and "Research on Key Technologies and Standards for Evolution of standard digitalization (Phase I)" will be launched respectively to solve the basic general and key technical problems of standard digitalization. Jointly carry out the research of standard digitization common technology, method standard, and platform tool. At the literature research level, Li Yuhua et al. [11] adopted the TOE framework and its extension field to model and analyze the digital transformation process under the influence of multiple factors. Yao Xinyi et al. [12] summarized the existing results of digital transformation and found that environment, organization, and technology are important factors affecting enterprise digital transformation. Ma Chao et al. [13] put forward relevant suggestions from three aspects: national strategy and international discourse power at the project level, pilot application and industrial collaboration in middle-level industries, core key technologies at the bottom level, and talent training. Overall, China has not yet established a clear path of standard digitalization, and all parties are still exploring this area. The factors affecting standard digitalization are diverse, and most existing studies have approached the topic from a single perspective [14–21], lacking a comprehensive and systematic analysis of the mechanisms, evaluation, and decision-making related to standard digitalization [22–31].

To address these gaps, we propose a novel Technology-Business-Organization-Environment (TBOE) maturity assessment model. The main contributions of this study are fourfold. First, we employ factor analysis to comprehensively identify macro and micro influencing factors of standard digitalization. Second, we construct the TBOE maturity assessment model using the Analytic Hierarchy Process (AHP) and the Delphi method, with explicit theoretical justification for extending the

traditional TOE framework by introducing Business as an independent dimension. Third, we establish a comprehensive indicator normalization procedure and provide empirically-grounded thresholds for stage division. Fourth, we conduct an empirical analysis using the electric power industry as a case study, including sensitivity analysis to validate model robustness.

The rest of this paper is organized as follows. Section 2 introduces Materials and methods, including influencing factors and mechanisms, methodology, empirical research and model validation. Section 3 introduces discussion and conclusions.

## Materials and methods

### Influencing factors and mechanisms

This section employs factor analysis to investigate the influencing factors and mechanisms of standard digitalization. The analysis consists of two steps: (1) Identifying influencing factors by comprehensively sorting the macro influencing factors and micro driving factors of standard digitalization, leading to the proposal of research hypotheses; and (2) Examining the mechanisms of action to form a theoretical framework, which lays the foundation for the indicator system and model framework.

**Macro influencing factors.** Macro factors focus on the external environment, including international, domestic (China), industrial, and other aspects. The details include:

(1) International development environment: Participation in international standardization organizations and activities, international standards development, etc., is a powerful breakthrough in strengthening the development layout of standard digitalization. Therefore, this paper proposes the following hypothesis:

H1: The international development environment exerts a significant positive influence on the standard digitalization.

(2) Domestic policy environment: The top-level design and policy guidance of digital standards at the national level can effectively drive the standard digitalization, reduce development obstacles, and improve the speed of transformation. Therefore, this paper proposes the following hypothesis:

H2: The domestic policy environment exerts a significant positive influence on the standard digitalization.

(3) Industry market environment: Industry-level standard digitalization construction degree, evolution route research, and related standard development progress, are important conditions and opportunities for standard digitalization. Therefore, this paper puts forward the hypothesis:

H3: The industry market environment exerts a significant positive influence on the standard digitalization.

**Micro driving factors.** Micro driving factors focus on the internal self, including technology, business, organization, and so on. The details include:

(1) Core key technologies: standard digitalization is essentially a change in organizational mode caused by technological change. The development of natural language processing(NLP), OCR, artificial intelligence generated content, and other technologies has changed the inherent form and application form of standards, which provides technical support and a resource base for the standard digitalization. Therefore, this paper proposes the following hypothesis:

H4: Core key technologies exerts a significant positive influence on the standard digitalization.

(2) New digital infrastructure: standard digitalization is based on the information network and standard data resource library as the base, opening up the interaction barriers between different systems, and improving the connectivity of standard data among various terminal devices, cloud platforms, and business systems are the supporting conditions for standard digitalization. Therefore, this paper proposes the following hypothesis:

H5: New digital infrastructure exerts a significant positive influence on the standard digitalization.

(3) Data security protection: Information and data security, intellectual property protection, etc. are emerging issues brought about by the standard digitalization, but also the bottom line that must be paid attention to. Therefore, this paper proposes the following hypothesis:

H6: Data security protection exerts a significant positive influence on the standard digitalization.

(4) Convergence of business scenarios: Promoting the integration of advanced digital technology and traditional business scenarios with standards can improve the service capability and chain stability of the standard digitalization. Therefore, this paper proposes the following hypothesis:

H7: The convergence of digital standards and business scenarios exerts a significant positive influence on the standard digitalization.

(5) Integration of new technology systems: The integration of digital standards and new technology systems is a key factor and innovation driver to promote the standard digitalization of the industry, and the two play a mutually reinforcing role. Therefore, this paper proposes the following hypothesis:

H8: The integration of digital standards and new technology systems exerts a significant positive influence on the standard digitalization.

(6) Integration of industrial engineering: The innovative integration of digital standards and major industrial engineering can promote the deepening of the innovation chain, standard chain, and industrial chain, and help the transformation and upgrading of the industry. Therefore, this paper proposes the following hypothesis:

H9: The integration of digital standards and major industrial engineering exerts a significant positive influence on the standard digitalization.

(7) Organization and management: The construction of an organizational management system with clear levels and clear responsibilities is the basis for the orderly promotion of standard digitalization. Therefore, this paper proposes the following hypothesis:

H10: Organization and management exerts a significant positive influence on the standard digitalization.

(8) Policy construction: Establishing and improving the supporting policies, management systems, and innovation incentive policies is the fundamental guarantee for the standard digitalization. Therefore, this paper proposes the following hypothesis:

H11: Policy construction exerts a significant positive influence on the standard digitalization.

(9) Talent cultivation: The construction of a talent cultivation and development incentive system, and the creation of a composite standard digital talent resource pool are the continuous driving forces of the standard digitalization. Therefore, this paper proposes the following hypothesis:

H12: Talent cultivation exerts a significant positive influence on the standard digitalization.

(10) Funding: Increasing investment in standard digital technology, scientific research projects, and financial support is an important support and financial guarantee for the standard digitalization. Therefore, this paper proposes the following hypothesis:

H13: Funding exerts a significant positive influence on the standard digitalization.

**Mechanism of action.** The TOE (Technology-Organization-Environment) framework is a comprehensive analysis framework based on the application scenarios of technology and is widely used in the analysis of influencing factors in the

process of new technology application and organizational transformation. Among them, the influence at the technical level mainly refers to the internal connection between technical characteristics and organizational structure. The influence at the organizational level mainly refers to the interaction between factors such as organizational structure and organizational management and the introduction of innovative technologies. The influence at the environmental level mainly refers to external conditions such as policy environment and industrial environment and the ecological environment.

The traditional TOE framework [32] has been widely applied to analyze technology adoption. However, in the context of standard digitalization, the TOE framework exhibits certain limitations. Specifically, the conventional TOE model overlooks the business-driven nature of standard digitalization. As shown in Fig 1, this study proposes the TBOE framework by explicitly introducing the Business dimension as an independent factor. The rationale is threefold: (1) Standard digitalization fundamentally aims to serve business scenarios—the integration of digital standards with business workflows is a core objective; (2) Business factors exhibit distinct characteristics that warrant independent analysis; (3) The separation of Business from Technology enables more precise identification of development priorities at different maturity stages, aligning with innovation diffusion patterns [33]. Thus, the TBOE model extends the theoretical boundary of TOE by repositioning business integration as an explicit analytical dimension.

DTF (Digital Transformation Framework) framework is a strategic evaluation model for digital transformation proposed by MattC et al. in 2015. It is applicable to the analysis of digital transformation in multiple fields under the current technological environment and consists of four elements. The first is the application of technology, which refers to the use of new technologies and the realization of their strategic goals. The second is value creation, which refers to the value impact of digital transformation on an organization. The third is the change in business structure, which refers to the transformation of business models caused by the realization of new technologies. The fourth is financial support, which refers to the financial conditions for applying new technologies to carry out digital transformation. It is the driving force for promoting changes in the first three aspects.

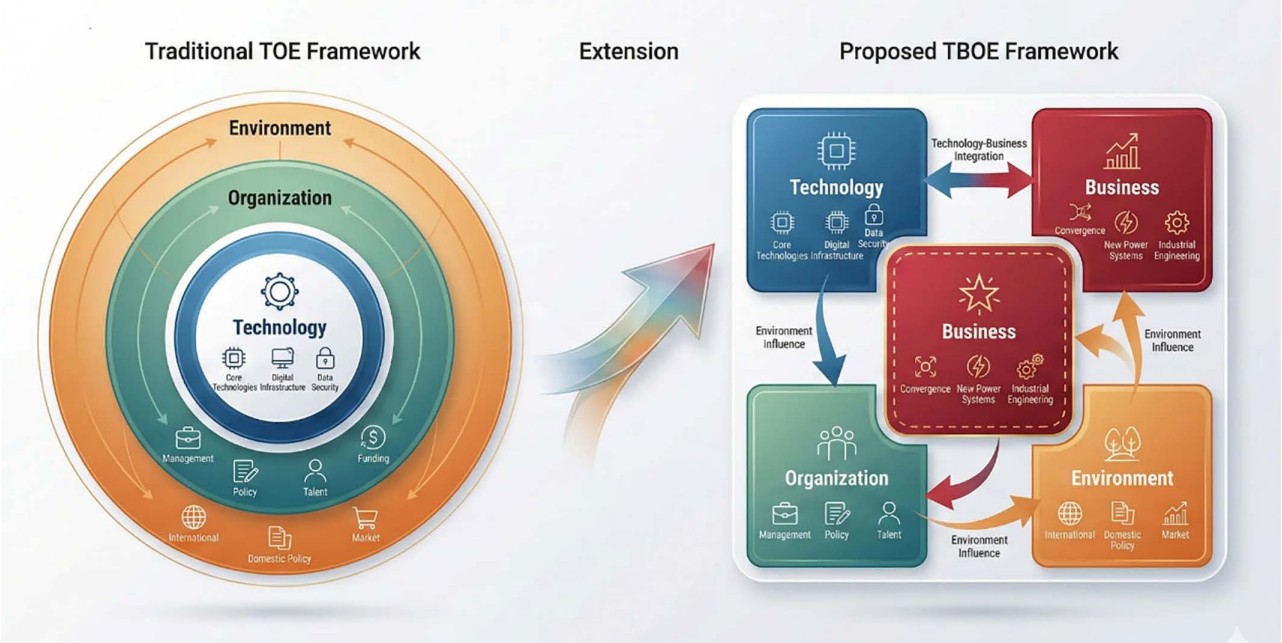

**Fig 1. The TBOE maturity assessment framework.**

Both of the above frameworks emphasize the impact of technology on digital transformation. The TOE framework focuses on the role of factors such as organizational management in digital transformation, while the DTF model pays attention to the changes in business models and organizational structures triggered during the digital transformation process. The financial support factors proposed in the DTF model are one aspect of the organizational-level influencing factors in the TOE framework. The TOE framework has the inclusiveness to accommodate multiple theories and be applicable to various scenarios in analyzing the influencing factors of new technology applications, but it needs to enhance targeted interpretations for specific business domains. The DTF model analyzes the particularity of digital technology, focusing on the structural changes and value creation brought about by the application of digital technology. However, it is necessary to strengthen the analysis of organizational changes within the field.

Therefore, in accordance with the characteristics of standard digitalization, this paper integrates the TOE framework and the DTF model, introduces the business factors of the DTF model into the TOE framework, expands the scope of the TOE framework, and constructs the TBOE analysis framework from four aspects: technical (T), business (B), organizational (O), and environmental (E). The relationship and mechanism among influencing factors are illustrated in Fig 2.

Technical aspect (T) is the original driver of the standard digitalization. standard digitalization is essentially a change in organizational model caused by technological change. The development of new technologies has changed the content form and application form of standards, provided a transformative force for industry innovation and development, and

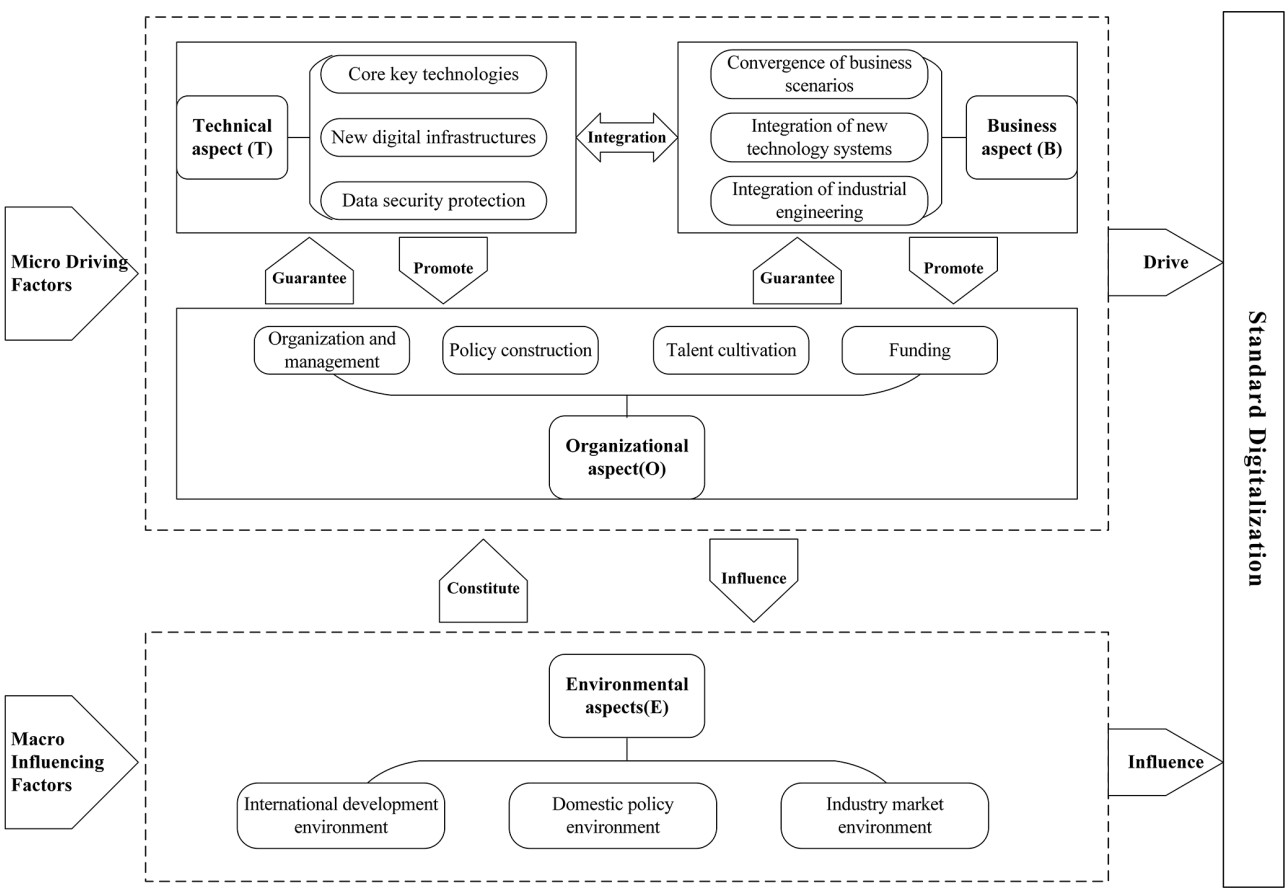

**Fig 2. Mechanism of influencing factors.**

provided technical support and resource base for the digital transformation of the standard, which will also directly drive corresponding changes in the business and organizational aspects.

The business aspect (B) is the ultimate destination of standard digitalization and is closely related to the technical aspect. In the process of the standard digitalization, accompanied by the deepening of technology application, the business structure will also undergo great changes, which are mainly reflected in the integration of digital standards and business scenarios, new technology systems, and industrial engineering. At the same time, the change in business structure also promotes the change in organizational form.

The organizational aspect (O) is the foundation of the standard digitalization. standard digitalization is a top-down change led by management, and the integration of technology and business levels also promotes changes at the organizational level, which can be reflected in organization and management, policy construction, talent cultivation, and funding. Meanwhile, the support from the organizational aspect also provides the core guarantee for the development of the technical and business aspects.

The environmental aspect (E) is a scenario variable for the standard digitalization. standard digitalization is influenced by domestic and international environments, including international, national, and industry level policies and market environments. Only by correctly facing the impact of environmental factors on the standard digitalization can we make full use of external support and mobilize internal resources to provide development soil for the standard digitalization.

## Methodology

Based on the analysis of the influencing factors and their mechanisms, this study utilizes the Analytic Hierarchy Process (AHP) to propose a TBOE maturity assessment model for standard digitalization. The Delphi method is employed to determine indicator weights, and the Likert scale method is used for index scoring.

**Justification for arithmetic mean method.** While some studies advocate for the geometric mean in synthesizing expert judgments [34], the arithmetic mean has been widely adopted when expert judgments are relatively consistent [35]. Forman and Peniwati [35] demonstrated that when the consistency ratio (CR) is low, the difference between arithmetic and geometric means becomes negligible. Given that our CR values are well below the 0.1 threshold (ranging from 0 to 0.055), the arithmetic mean provides a reasonable approximation.

Given the heterogeneous nature of measurement scales across indicators (Table 1), normalization is essential. This study adopts the min-max normalization method:

$$V_{normalized} = (V_{original} - V_{min}) / (V_{max} - V_{min})$$

(1)

**Determine the index system.** Under the framework of TBOE, a multi-level structure model is constructed, and a maturity assessment index system for the standard digitalization is proposed. It includes four first-level indicators and 13 second-level indicators. The index system is shown in Fig 3.

**Invite experts to distribute questionnaires.** Select experts and experienced staff in the field of standardization digitization and industry to form an expert consulting team. Develop a scoring table for expert consultation, and invite experts to use the "1-9 scale method" to score the standard digital transformation influencing factors extracted in this paper according to the grade importance coefficient provided.

**Data processing and construct a judgment matrix.** After the expert questionnaires were collected, the arithmetic mean method was used to process the valid questionnaires and construct the judgment matrix needed for the research. If there are n compared elements, a comparison judgment matrix H of order n*n is obtained, as shown in Eq. (1):

$$H = \begin{bmatrix} u_{11} & \cdots & u_{1n} \\ \vdots & \ddots & \vdots \\ u_{n1} & \cdots & u_{nn} \end{bmatrix}$$

(2)

**Table 1. Normalization parameters for each indicator.**

| Indicator | Original Scale | $V_{min}$ | $V_{max}$ | Normalization |
|---|---|---|---|---|
| T1 | TRL 1–9 × 10 items | 10 | 90 | (V-10)/80 |
| T2 | Count (0–7 domains) | 0 | 7 | V/7 |
| T3 | Maturity 1–5 × 6 items | 6 | 30 | (V-6)/24 |
| B1 | Count (0–26 scenarios) | 0 | 26 | V/26 |
| B2 | Count (0–10 systems) | 0 | 10 | V/10 |
| B3 | Count (0–5 projects) | 0 | 5 | V/5 |
| O1-O3 | Binary (0/1) | 0 | 1 | V (unchanged) |
| O4 | Percentage (%) | 0 | 100 | V/100 |
| E1 | Count (0–5 activities) | 0 | 5 | V/5 |
| E2 | Count (0–3 policies) | 0 | 3 | V/3 |
| E3 | Count (0–3 items) | 0 | 3 | V/3 |

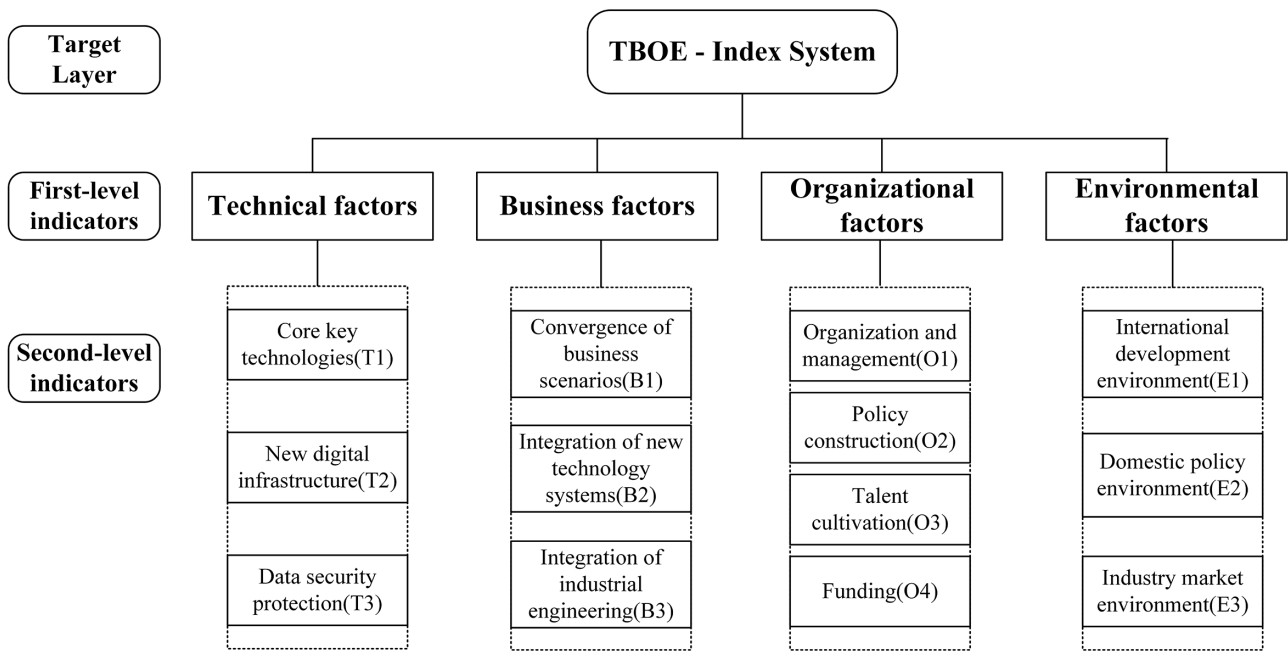

**Fig 3. The proposed TBOE index system.**

Among them, uij represents the numerical value of the relative importance of the element ui to the element uj (i = 1, 2,……, n; j = 1,2,……,n).

**Hierarchical single sorting and consistency check.** For the judgment matrix H, the eigenvector corresponding to the characteristic root satisfying HW = $\lambda_{max}$ W is calculated as the importance order of each element, and the weight is assigned to Wi after standardization. The weight of each index at the same level is determined, and the eigenvector corresponding to its maximum characteristic root is obtained according to the judgment matrix, and the weight is obtained after normalization. The consistency check is performed using the following equation:

$$CR = \frac{CI}{RI}$$

(3)

where CR represents the random consistency ratio of the judgment matrix, CI represents the consistency index of the judgment matrix, and RI is the average random consistency index of the judgment matrix. The solution formula is shown in Eq. (4):

$$CI = (\lambda_{max} - n)/(n - 1)$$

(4)

where $\lambda_{max}$ is the largest eigenvalue of the judgment matrix, and n is the order of the judgment matrix. After calculation, if CR < 0.1, it is considered that the consistency of the judgment matrix is acceptable, that is, it has passed the consistency test. If CR ≥ 0.1, the judgment matrix should be revised until it passes the consistency test.

**Hierarchical total sorting and consistency checking.** The calculation method of total hierarchical ranking and consistency test is the same as the weight determination method of single hierarchical ranking, and the total ranking of each index relative to the target layer is obtained. Then, the consistency and randomness of the weights are tested to determine whether the weights are reasonable.

The consistency test of the total ranking is calculated according to Eq. (5):

$$CR = \frac{\sum_j w_{cj} CI_j}{\sum_j w_{cj} RI_j} < 0.1$$

(5)

where $CI_j$ is the single ranking consistency index relative to the j-th element of the criterion layer, and $RI_j$ is the corresponding average random consistency index. If CR < 0.1, the total ranking of the hierarchy passes the consistency test; otherwise, the values of the elements of the judgment matrix need to be adjusted.

**Assessment of transition stage.** According to the life cycle theory, the Gartner emerging technology maturity curve [36], combined with the development characteristics of standard digitalization, the standard digitalization is divided into three stages: research & development stage, implementation stage, and promotion stage.

Considering the evaluation index system constructed above, after determining the scores and weight distribution of each index, a comprehensive maturity assessment model is proposed in order to evaluate the stage of standard digitalization and assist comprehensive decision-making, as shown in Eq. (6):

$$L_{TBOE} = f(T, B, O, E) = \sum T_i W_{Ti} + \sum B_i W_{Bi} + \sum O_i W_{Oi} + \sum E_i W_{Ei}$$

(6)

where LTBOE stands for the comprehensive evaluation value of the standard digitalization. $T_i$ and $W_{Ti}$, $B_i$ and $W_{Bi}$, $O_i$ and $W_{Oi}$, and $E_i$ and $W_{Ei}$ are the index scores and weights of the indicators normalized at the technical aspect, the business aspect, the organizational aspect, and the environmental aspect, respectively.

According to the area integral proportion of each stage under the Gartner emerging technology maturity curve, the stage division criteria of the standard digitalization is proposed. According to the comprehensive evaluation value $L_{TBOE}$ calculated by Eq. (6), the corresponding stage of the standard digitalization can be identified, as shown in Table 2. The stage division thresholds (0.3 and 0.8) are derived from the Gartner Hype Cycle framework [36] combined with the

Table 2. Stage division criteria of the standard digitalization.

| $L_{TBOE}$ Comprehensive Evaluation Value | Stage of standard digitalization |
|---|---|
| $0 \le L_{TBOE} < 0.3$ | Research&development stage |
| $0.3 \le L_{TBOE} < 0.8$ | Implementation stage |
| $0.8 \le L_{TBOE} \le 1$ | Promotion stage |

"30-50-20" rule. According to Linden and Fenn [36], the five phases correspond to different proportions of the technology adoption process. These thresholds are further supported by Rogers' Diffusion of Innovations theory [33]. During Delphi consultation, experts recommended 0.28 (SD = 0.05) for the R&D-Implementation boundary and 0.79 (SD = 0.08) for the Implementation-Promotion boundary, rounded to 0.3 and 0.8.

(1) Research & development stage of the standard digitalization. It ranges from the technical research of standard text structure to the digital standard intelligence development and intelligent service of machine language state. This stage is technology-intensive and requires a lot of investment in infrastructure and technology research and development.

(2) Implementation stage of the standard digitalization. It begins with the generation of digital standards for machine language state and ends with the application of digital standards and business integration. This stage is experience-intensive and labor-intensive, requiring a large amount of industry experience, personnel, and capital investment.

(3) Promotion stage of the standard digitalization. Starting from the application and normal operation of digital standards and services integration, it is a long-term stage. This stage is resource-intensive and requires a large number of national and international standardization activities and resource investment.

**Empirical research and model validation**

**Selection of scenario and index.** Based on the above research hypothesis and the TBOE maturity assessment model for standard digitalization construction, the empirical research and model validation is carried out by taking the electric power industry as an example. Combined with the characteristics of the electric power industry, the second-level indicators are further concretized as the measurement index of the electric power industry, and the basis for assigning the index is determined, as shown in Table 3.

**Model validation of the electric power industry.**

(1) Expert selection and questionnaire distribution

An expert advisory panel was formed by inviting specialists from the fields of standardization, digitization, and the power industry. Questionnaires were distributed through both online and offline channels. Experts were selected through three primary channels: 1) Standardization experts: Questionnaires were distributed through the Science and Technology Management Department to all researchers at the China National Institute of Standardization; 2) Digitalization experts: Targeted distribution was conducted to 85 member units of the National Standardization Working Group (SAC/SWG 29) and 10 project units under the National Key R&D Program "Key Technologies and Standards for Digital Evolution of Standards (Phase I)"; 3) Power industry experts: Questionnaires were distributed to relevant entities including the project team of State Grid Corporation's "Research on Implementation Pathways and Key Technologies for Standardization Digitalization", State Grid Information & Telecommunication Industry Group Co., Ltd., and China Electric Power Research Institute. The full questionnaire is shown in Supporting information (S1 File).

Ultimately, 84 valid expert questionnaires were collected. The average age of the experts was 46.42 years, and 60.71% held senior professional titles. The experts were affiliated with institutions including China National Institute of Standardization, China Electronics Technology Standardization Institute, Zhejiang University, Zhijiang Laboratory, Shandong Provincial Computing Center (National Supercomputing Center in Jinan), State Grid Information and Communication Industry Group Co., LTD., China Electric Power Research Institute, State Grid Zhejiang Electric Power Co., LTD., Xi 'an Jiaotong University and other institutions.

(2) Data processing and Judgment Matrix Construction

The 84 collected expert scoring forms were processed using the arithmetic mean method to establish judgment matrices for each level. For a given judgment matrix H, the eigenvalue and weights were calculated using the root method, with the following specific steps:

**Table 3. Measurement index and assignment basis of the electric power industry.**

| First-level indicators | Second-level indicators | Measurement index | Assignment basis |
|---|---|---|---|
| Technical factors (T) | T1:Core key technologies | Technology Readiness Level (TRL) | With reference to the Technology Readiness Level (TRL) proposed by NASA in 1969, the technology maturity level of 10 core key technologies of the electric power industry (NLP, OCR, ontology modeling, big data, life cycle management, artificial intelligence, and large model, knowledge graph, cloud computing, visual analysis, knowledge reasoning) is assigned 1–9 values. The sum of scores is the total score of the index. |
| | T2:New digital infrastructure | Number of domains covered by the standard data base | With reference to the White Paper on standard digitalization of State Grid Corporation released in March 2024, 0/1 is assigned according to whether the construction of standard data resource library is completed in seven fields of electric power industry (low-carbon electricity, equipment management, power load management, information network security, infrastructure management, supply chain management, and dispatch operation), and the sum of scores is the total score of the index. |
| | T3:Data security protection | Standard data security capability maturity | According to the national standard GB/T 37988–2019 "Information Security Technology Data Security Capability Maturity Model" of the People's Republic of China, the data security process dimension is composed of six aspects: data acquisition, data transmission, data storage, data processing, data exchange, and data destruction. The maturity of the electric power industry standard data security capability is assigned 1–5 values, and the sum of scores is the total score of the index. |
| Business factors (B) | B1:Convergence of business scenarios | Integration degree between digital standards and traditional business scenarios of electric power | With reference to the White Paper on standard digitalization of State Grid Corporation released in March 2024,according to 26 business scenarios of the electric power industry (low-carbon application of industrial carbon efficiency code, low-carbon application of living carbon master, application of corporate carbon management planning, digitization of carbon monitoring standards, corporate carbon inventory and carbon asset management, optimization and management of corporate double carbon evaluation indicators, whole process technical supervision, intelligent patrol of unattended substation, oil-immersed power transformer insulation test, transmission line UAV inspection, load Standardization of resource mapping, standardization of accurate management of air conditioning load, standardization of virtual power plant, information system security testing, evaluation of red and blue line indicators of information system, difference comparison of information network security standards, environmental and water protection design and review, quality control of electrical equipment handover test, three-dimensional design compliance review, procurement standards application, material quality detection and control application, power logistics Standard application, standard numerical assistant for all business scenarios, distribution network scheduling management, production safety assurance capability assessment, and network-related safety inspection) are integrated with the digital standards of the electric power industry. The 0/1 value is assigned, and the sum of scores is the total score of the index. |
| | B2:Integration of new technology systems | Integration of digital standards with new power systems | With reference to the "Several Opinions on Accelerating the Development of Digital and Intelligent Energy" issued by the National Energy Administration of China in March 2023, the comprehensive scoring system has been adopted. According to the seven types of scenarios of the new power system (power system multi-energy complementary joint dispatching, power grid transient stability intelligent assessment and early warning, substation and converter station intelligent operation and inspection, transmission line intelligent inspection, distribution intelligent operation and maintenance, power grid disaster intelligent perception, electric carbon measurement, and accounting monitoring) are integrated with the digital standards of the electric power industry. The sum of scores is the total score of the index. |
| | B3:Integration of industrial engineering | Integration degree of electric power digital standards and major industrial projects | With reference to the Development Outline of Technical Standards of the State Grid Co., LTD., the comprehensive scoring system is adopted to assign 0/1 values according to whether the five major projects of industrial digitalization (UHV, new energy, emergency management, rural revitalization, and smart supply chain) are integrated with the digital standards of the electric power industry, and the sum of scores is the total score of the index. |

*(Continued)*

**Table 3.** (Continued)

| First-level indicators | Second-level indicators | Measurement index | Assignment basis |
|---|---|---|---|
| Organizational factors (O) | O1:Organization and management | Construction of the organization management system for the standard digitalization | According to whether to establish the relevant organization and management department of the standard digitalization of the electric power industry and establish an effective coordination mechanism, the index is assigned a 0/1 value. |
| | O2:Policy construction | Construction of supporting systems for the standard digitalization | According to whether the related policies, management systems, and scientific and technological innovation supporting systems of the electric power industry standard digitalization are formulated, the index is assigned a 0/1 value. |
| | O3:Talent cultivation | The construction of personnel training mechanism for the standard digitalization | The index was assigned a value of 0/1 based on whether to establish a talent recruitment, training policy, and development incentive mechanism for the standard digitalization in the electric power industry. |
| | O4:Funding | Proportion of R&D expenditure | Proportion of R&D expenditure＝R&D expenditure of electric power industry/ total cost expenditure ×100%, the average value of the recent three years is taken as the index score. |
| Environmental factors (E) | E1:International development environment | Participation in international standardization activities | Participation of electric power industry experts in international standardization activities (held key positions in international and regional standardization organizations; relevant international standard proposal project; relevant new technology field proposal project; Establishment of relevant multilateral and bilateral cooperation projects; relevant national or industry standards are adopted or modified by other countries/regions) for 0/1 assignment, and the sum of scores is the total score of the index. |
| | E2:Domestic policy environment | Support degree of national policies and regulations | Degree of support for policies and regulations related to the digitalization of national standards (relevant policy documents issued by the state; the state promulgates relevant laws and regulations; the state starts relevant the National Key R&D Program)to assign 0/1, and the sum of scores is the total score of the index. |
| | E3:Industry market environment | Construction degree of standard digitalization in the electric power industry | The construction degree of the standard digitalization in the electric power industry (establishment of the working group of the standard digitalization in the electric power industry; publishing relevant standard digitalization of the electric power industry; starting key projects related to standard digitalization in the electric power industry) for 0/1 assignment, and the sum of scores is the total score of the index. |

a) Calculate the product of elements in each row of matrix H:

$$M = \begin{bmatrix} M_1 \\ M_2 \\ \vdots \\ M_3 \end{bmatrix} = \begin{bmatrix} u_{11} \times u_{12} \times \cdots \times u_{1n} \\ u_{21} \times u_{22} \times \cdots \times u_{2n} \\ \vdots \\ u_{n1} \times u_{n2} \times \cdots \times u_{nn} \end{bmatrix} \tag{7}$$

b) Find the n-th root of M and normalize it to get the weight:

$$w_i = \frac{\overline{M_i}}{\sum_i \overline{M_i}} \tag{8}$$

c) Find the maximum eigenvalue of H:

$$\lambda_{max} = \frac{1}{n}\sum_{i=1}^{n}\frac{(Hw)_i}{w_i}$$

(9)

(3) Hierarchical single sorting and consistency check

Following the method described above, the expert scoring data was statistically processed to establish judgment matrices at each level. The weights and consistency test results for all levels of indicators are presented in Table 4.

(4) Hierarchical total sorting and consistency checking

The total ranking of each influencing factor in this model was calculated, and a consistency check was performed. The result, CR = 0.055 < 0.1, indicates that the model passed the consistency test and is valid.

By calculating the weights of the indicators, the weights and ranking of the factors influencing the digitization of standards in the electric power industry are obtained, as shown in Fig 4.

**Sensitivity analysis.** To assess the robustness of the model results, we conducted sensitivity analysis on two dimensions: (1) weight perturbation analysis, and (2) indicator score uncertainty analysis.

We examined the stability of the stage classification by perturbing the first-level indicator weights by ±10% and ±20%, as shown in Table 5.

Under all weight perturbation scenarios (±20%), $L_{TBOE}$ values range from 0.486 to 0.518, remaining within the Implementation stage (0.3–0.8).

Monte Carlo simulation (1,000 iterations, ± 15% perturbation) shows $L_{TBOE}$ follows a normal distribution with mean = 0.498 and SD = 0.042. The 95% confidence interval [0.416, 0.580] lies entirely within the Implementation stage. None of the simulations resulted in a stage classification change.

**Innovation model and path planning.** Based on the TBOE maturity assessment model, the comprehensive assessment value of the power industry $L_{TBOE}$ is 0.502 according to Eq. (6). Therefore, the electric power industry is in the early-to-middle implementation stage of the standard digitalization, as shown in Fig 5.

In terms of innovation mode, referring to the theory of technological innovation process and according to the system integration concept of "four-in-one" in the planning and design link, equipment management link, construction and operation link, and service extension link of the electric power industry, four stages of innovation mode are designed respectively: Standard collaborative development mode, standard machine application mode, standard process embedding mode, and standard intelligent service mode.

In the link of power grid planning and design, the innovation mode focuses on breaking the limitation of the standard using paper text as the carrier and strengthening the interconnection capability, dynamic updating capability, and collaborative development capability of relevant parties of the standard. In the electric power equipment management link, the focus of the innovation mode is to transfer the standard to the sensing equipment in the form of data, to achieve scenarios such as UAV inspection, substation intelligent inspection, and equipment status monitoring, and to promote the construction of a new power system. In the link of electric power grid construction and operation, the focus of the innovation mode is to use digital standards through all aspects of power grid construction such as scheme design, bidding, construction, commissioning, settlement, demolition, etc., to achieve strong integration of power grid digital standards and business, and continue to improve the level of power grid construction. In the power grid service extension link, the focus of the innovation model is to achieve unified data services, unified business services, and unified technical services, to build a standardized intelligent service platform for effective integration with measurement, inspection and testing, certification and accreditation, and other links, in a timely, effective and intelligent manner to provide standardized services. Organizational factors and environmental factors, as secondary influencing factors, exist in the form of necessary external support in the construction of the digital innovation model of the electric power industry standard.

**Table 4. Weight and consistency test results of all levels of indicators.**

| First-level indicators | Factors | T | B | O | E |
|---|---|---|---|---|---|
| | T | 1 | 2.1 | 7.2 | 6.2 |
| | B | 0.467 | 1 | 3.6 | 3.1 |
| | O | 0.139 | 0.278 | 1 | 2.5 |
| | E | 0.161 | 0.323 | 0.4 | 1 |
| | Weight | 0.559 | 0.273 | 0.1 | 0.068 |
| | $\lambda_{max}$ =4.146, CI=0.049, RI=0.882<br>CR=CI/RI=0.055<0.1. Consistency check passed. | | | | |
| Second-level indicators-Technical aspect | Factors | T1 | | T2 | T3 |
| | T1 | 1 | | 2.222 | 4.348 |
| | T2 | 0.45 | | 1 | 1.923 |
| | T3 | 0.23 | | 0.52 | 1 |
| | Weight | 0.596 | | 0.267 | 0.138 |
| | $\lambda_{max}$ =3, CI=0, RI=0.525<br>CR=CI/RI=0<0.1. Consistency check passed. | | | | |
| Second-level indicators-Business aspect | Factors | B1 | | B2 | B3 |
| | B1 | 1 | | 2.23 | 7.73 |
| | B2 | 0.448 | | 1 | 3.81 |
| | B3 | 0.129 | | 0.262 | 1 |
| | Weight | 0.630 | | 0.291 | 0.079 |
| | $\lambda_{max}$ =3.001, CI=0, RI=0.525<br>CR=CI/RI=0.001<0.1. Consistency check passed. | | | | |
| Second-level indicators-Organizational aspect | Factors | O1 | O2 | O3 | O4 |
| | O1 | 1 | 0.391 | 0.163 | 0.157 |
| | O2 | 2.56 | 1 | 0.459 | 0.431 |
| | O3 | 6.13 | 2.18 | 1 | 0.442 |
| | O4 | 6.35 | 2.32 | 2.26 | 1 |
| | Weight | 0.062 | 0.165 | 0.304 | 0.469 |
| | $\lambda_{max}$ =4.075, CI= 0.025, RI= 0.882<br>CR=CI/RI=0.028<0.1. Consistency check passed. | | | | |
| Second-level indicators-Environmental aspect | Factors | E1 | | E2 | E3 |
| | E1 | 1 | | 0.287 | 0.14 |
| | E2 | 3.48 | | 1 | 0.452 |
| | E3 | 7.13 | | 2.21 | 1 |
| | Weight | 0.085 | | 0.290 | 0.625 |
| | $\lambda_{max}$ =3.001, CI=0, RI= 0.525<br>CR=CI/RI=0.001<0.1. Consistency check passed. | | | | |

In terms of path planning, the standard digitalization in the electric power industry needs to adhere to the overall principle of "technology-driven, business-oriented, organizational optimization, and policy guidance", and carry out path planning from the three stages of standard digitalization around the goal of taking data as the core element and releasing the value of standard data assets. In the research & development stage of the standard digitalization, it is necessary to adhere to the technology-driven path, which can be divided into three directions: the standard digitalization content, the standard digitalization development, and the standard digitalization application, and it is necessary to strengthen key technology research, the construction of electric power industry standard data resources and data security protection. In the implementation stage of standard digitalization, it is necessary to adhere to the business-oriented and organizational

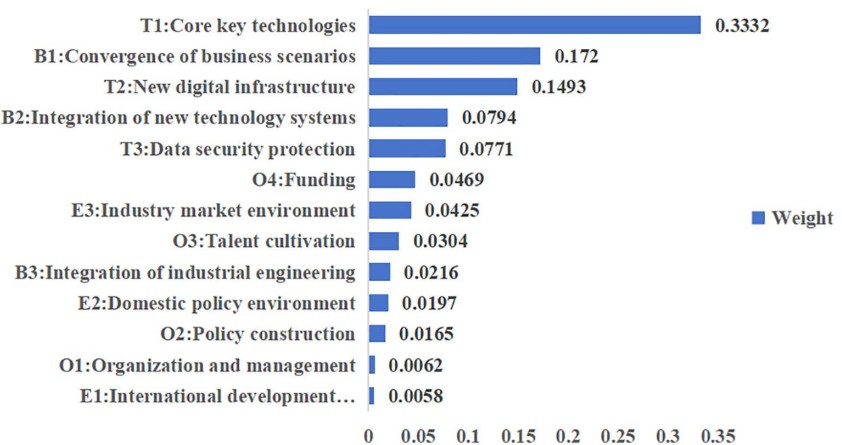

**Fig 4. Ranking of influencing factors.**

**Table 5. Weight sensitivity analysis results.**

| Scenario | T | B | O | E | $L_{TBOE}$ | Stage |
|---|---|---|---|---|---|---|
| Baseline | 0.559 | 0.273 | 0.100 | 0.068 | 0.502 | Implementation |
| T+20% | 0.671 | 0.218 | 0.067 | 0.044 | 0.518 | Implementation |
| T −20% | 0.447 | 0.367 | 0.113 | 0.073 | 0.486 | Implementation |
| B+20% | 0.502 | 0.328 | 0.103 | 0.067 | 0.495 | Implementation |
| B −20% | 0.612 | 0.218 | 0.102 | 0.068 | 0.509 | Implementation |
| O+20% | 0.546 | 0.266 | 0.120 | 0.068 | 0.498 | Implementation |
| E+20% | 0.549 | 0.268 | 0.101 | 0.082 | 0.501 | Implementation |

optimization path, focus on the integration of digital standards with traditional power grid business scenarios, new power systems, and major industrial projects, and promote the deep integration and application of new digital technologies and power grid business with standards to help power system transformation and upgrading. At the same time, it should focus on the organization management system, supporting management system, talent training mechanism, and research and development fund investment to provide the core guarantee for the standard digitalization in the electric power industry. In the promotion stage of standard digitalization, it is necessary to adhere to the policy guidance path, comply with and grasp the international, domestic, and industry development situation, timely formulate matching strategies and plans, accelerate the formation and development of the standard digitalization, and move with the trend.

## Discussion and conclusions

### Comparison with existing studies

To position our contribution within the broader literature, we systematically compare the TBOE model with existing studies on technology adoption and digital transformation frameworks. Table 6 presents a comparative analysis of representative studies.

Our TBOE model makes four major theoretical contributions to the existing body of knowledge:

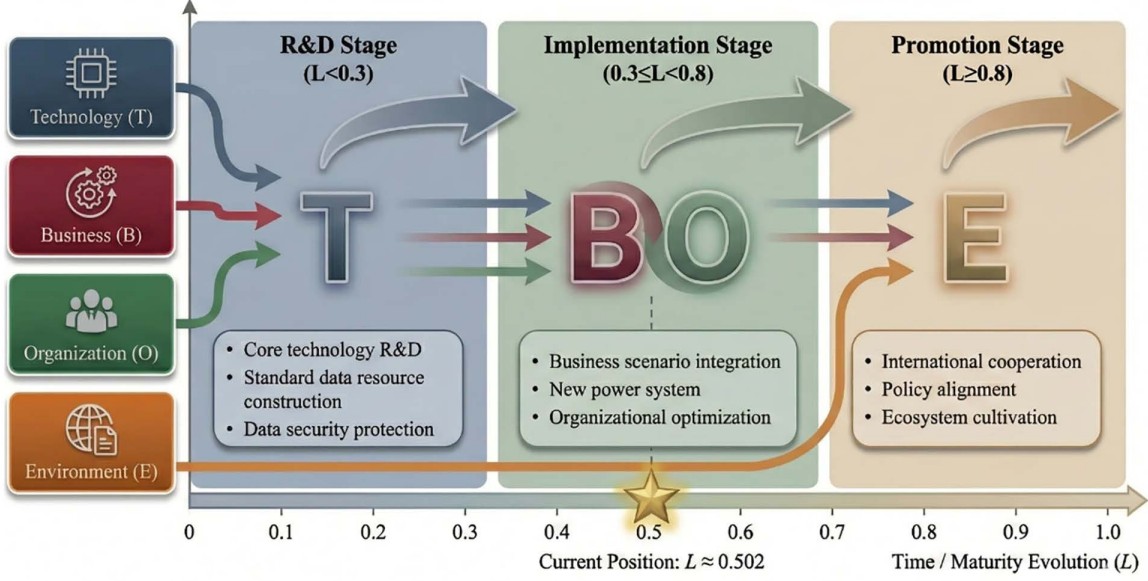

**Fig 5. Path planning framework for standard digitalization in the power industry.**

**First, theoretical extension of the TOE framework:** While the traditional TOE framework [32] and its extensions [37–39] have been widely applied to technology adoption studies, they typically subsume business-related factors under the Technology or Organization dimensions. Our TBOE model explicitly introduces Business as an independent fourth dimension, recognizing that standard digitalization is fundamentally business-driven. This extension addresses a critical gap identified by Li et al. [11] who noted that existing frameworks inadequately capture the business transformation aspects of digitalization.

**Second, quantitative maturity assessment methodology:** Unlike prior studies that primarily employ binary adoption measures [37] or qualitative configuration analysis [11], our model provides a continuous, quantitative assessment of digitalization maturity. The integration of AHP with Delphi method enables systematic weight determination with consistency verification, addressing the methodological limitation of subjective weighting in existing maturity models.

**Third, theoretically-grounded stage division:** Previous digital transformation studies often lack clear criteria for stage classification. Our model derives stage division thresholds (0.3 and 0.8) from established theories—the Gartner Hype Cycle [36] and Rogers' Diffusion of Innovations [33]—and validates them through expert consultation. This provides a replicable framework for assessing digitalization progress across different contexts.

**Fourth, domain-specific empirical validation:** While many TOE studies remain at the conceptual level [39] or focus on general IT adoption, our study provides rigorous empirical validation within the electric power industry. The sensitivity analysis further demonstrates the robustness of our findings, addressing concerns about model stability that are rarely examined in existing studies.

In summary, the TBOE model advances the theoretical frontier by extending the TOE framework, providing quantitative assessment capabilities, and offering empirically-validated stage classification. These contributions address key limitations in existing digital transformation research and provide a foundation for future studies in standard digitalization across various industries.

## Conclusions

This study investigated the influencing factors and mechanisms of standard digitalization. Based on this investigation, a TBOE maturity assessment model was constructed. Integrating lifecycle theory, the process of standard digitalization was

**Table 6. Comparison with existing studies.**

| Study | Framework | Methodology | Domain | Key Limitation |
|---|---|---|---|---|
| Tornatzky & Fleischer (1990) [32] | TOE (3 dimensions) | Conceptual | General technology adoption | No quantitative assessment; Business subsumed under Technology |
| Zhu et al. (2006) [37] | Extended TOE | Survey + Regression | E-business adoption | Binary adoption measure; No maturity stages |
| Li et al. (2019) [11] | TOE | fsQCA | Enterprise digital transformation | Configuration paths only; No comprehensive scoring |
| Oliveira & Martins (2011) [38] | TOE Review | Literature review | IT adoption models | Descriptive synthesis; No new framework |
| Baker (2012) [39] | TOE Analysis | Theoretical | IS theory building | Conceptual discussion; No empirical validation |
| **This Study** | **TBOE (4 dimensions)** | **AHP + Delphi + Sensitivity Analysis** | **Standard digitalization** | **Industry-specific; Requires adaptation for other sectors** |

divided into three stages: research and development, implementation, and promotion; a comprehensive grading model was further proposed. Utilizing the established indicator system, the electric power industry served as a case study to validate the effectiveness of the model and method through empirical analysis. Based on the model results, the current stage of standard digitalization in the power industry was evaluated, and corresponding innovation models and path planning strategies were proposed.

The TBOE model developed in this study is a universal theoretical framework applicable to various industries or enterprises for assessing standard digitalization capability maturity, determining capability levels, and thereby informing subsequent path planning. However, when applying the model across different industries, the specific measurement indicators and scoring bases will vary. It is necessary to design tailored measurement indicators and scoring bases under the secondary indicators proposed herein, based on the unique characteristics of each industry. Among these, the indicators for the technical, organizational, and environmental factors can generally be reused directly. In contrast, the indicators under business factors will differ significantly across industries and must be designed based on the specific context of the industry itself.

China's standard digitalization is still in its nascent stages. It requires policy guidance at the national level to clarify the strategic positioning of standard digitalization development, strengthen top-level design and strategic research, and plan the pathway for standard digitalization. These steps are crucial to further improve the working mechanism and provide greater support for standard digitalization initiatives. Finally, we propose the following three recommendations:

First, seize the opportunities presented by digitization and unlock the value of standard data. In the digital economy era, standards constitute vital data assets. It is necessary to build digital standard platforms and upgrade production tools to fully leverage the value of standards. Relying on standard digitalization can drive business and business model innovation, releasing greater efficiency.

Second, maintain a focus on innovation and overcome the core technologies underpinning standard digitalization. Systematically conduct research on common and key technologies, focusing on resolving fundamental common issues such as relevant theories, methods, and models, as well as key technical challenges like semantic recognition, data analysis, and rule integration in field applications. Utilize digital technologies to empower both the standards themselves and their entire lifecycle.

Third, adopt a comprehensive and systematic approach to cultivate an industry application ecosystem for standard digitalization. Conduct in-depth research tailored to industrial needs, expand the scope of pilot projects (e.g., machine-readable standards), and systematically promote application practices. Based on the characteristics and specific needs of different industries, research technologies, models, methods, tools, and data sets suitable for industrial applications.

This study has several limitations: (1) The empirical validation was conducted solely within the electric power industry in China—generalizability to other industries requires further investigation; (2) The stage division thresholds may benefit from refinement as more empirical data becomes available; (3) The cross-sectional nature does not track longitudinal changes. Future research could address these by conducting multi-industry comparative studies, performing longitudinal studies, and developing industry-specific threshold calibrations.

## Supporting information

**S1 File. AHP expert consultation scoring table.**
(DOCX)

## Author contributions

**Conceptualization:** Xize Liu, Yiyi Wang.

**Data curation:** Bingyan Zhang, Xinrui Hu.

**Formal analysis:** Nana Niu.

**Funding acquisition:** Xize Liu.

**Investigation:** Nana Niu, Xinrui Hu.

**Project administration:** Xize Liu, Yiyi Wang.

**Resources:** Yiyi Wang, Jingsheng Li.

**Software:** Jingsheng Li.

**Supervision:** Yiyi Wang.

**Validation:** Nana Niu, Jingsheng Li.

**Writing – original draft:** Xize Liu, Bingyan Zhang, Xinrui Hu.

**Writing – review & editing:** Yiyi Wang.

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
