## [Decision Letter · Decision Letter 0]

9 Jun 2025

PONE-D-24-60444TBOE Maturity Assessment Model for Digitalization of Standards ： An Empirical Analysis Using AHP and Delphi MethodPLOS ONE

Dear Dr. Zhang,

Thank you for submitting your manuscript to PLOS ONE. After careful consideration, we feel that it has merit but does not fully meet PLOS ONE’s publication criteria as it currently stands. Therefore, we invite you to submit a revised version of the manuscript that addresses the points raised during the review process.

Although one (several) reviewers recommended rejecting this manuscript, I believe it can still be considered with significant revisions. Please consider reviewer's comments carefully.

We look forward to receiving your revised manuscript.

Kind regards,

Nhat-Luong Nhieu, Ph.D.

Academic Editor

PLOS ONE

Journal Requirements:

2. Please remove all personal information, ensure that the data shared are in accordance with participant consent, and re-upload a fully anonymized data set.

Additional Editor Comments:

Although one (several) reviewers recommended rejecting this manuscript, I believe it can still be considered with significant revisions. Please consider reviewer's comments carefully.

Reviewers' comments:

Reviewer's Responses to Questions

**Comments to the Author**

1. Is the manuscript technically sound, and do the data support the conclusions?

Reviewer #1: Yes

Reviewer #2: No

2. Has the statistical analysis been performed appropriately and rigorously? 

Reviewer #1: Yes

Reviewer #2: No

3. Have the authors made all data underlying the findings in their manuscript fully available?

Reviewer #1: Yes

Reviewer #2: No

4. Is the manuscript presented in an intelligible fashion and written in standard English?

Reviewer #1: Yes

Reviewer #2: No

5. Review Comments to the Author

Reviewer #1: This paper presents a framework for assessing the digitalization of standards using the TBOE (Technology-Business-Organization-Environment) model, supported by the AHP and Delphi methods. The authors also apply the proposed framework to a case study in China’s electric power industry. While the topic is both interesting and highly relevant, the paper requires some revisions:

1- The authors state that they integrated the TOE and DTF frameworks to construct the TBOE model; however, they do not provide sufficient explanation of these original frameworks. The manuscript should include a clear description of both TOE and DTF, along with a discussion of the existing literature that justifies the need for TBOE as an extension and highlights its added value.

2- The electric power industry is a compelling case study; however, the authors should discuss the model’s applicability to other industries and provide suggestions for future validation across different sectors.

3- Methodologically, the use of AHP and the Delphi method is appropriate. However, the authors need to provide more details regarding the selection criteria for experts, their backgrounds, and the full questionnaire used. The authors should include the full questionnaire as an appendix or supplementary file.

4- The authors do not provide sufficient explanation of how the Delphi method was applied in this study. A brief overview of the Delphi method should be included, along with details such as the number of rounds conducted, how consensus was measured, and how expert feedback was processed.

5- The figures in the manuscript should be drawn again with clearer labels and improved resolution to enhance presentation quality.

Reviewer #2: Dear Authors,

Thank you for the opportunity to review your manuscript, which proposes the TBOE Maturity Assessment Model for evaluating the digitalization of standards. The paper addresses a timely and practically relevant topic—particularly as digital transformation efforts expand across industries. Your empirical focus on the electric power sector and the application of AHP and Delphi methods are commendable efforts.

However, in its current form, the manuscript falls short of the publication criteria for this journal. I therefore recommend rejection at this time. That said, I believe this work has the potential to be resubmitted elsewhere after careful revision. I hope the following detailed comments will assist you in improving the manuscript for possible submission to another journal.

1. Theoretical Justification and Model Positioning

While the extension from TOE to TBOE is logical, the manuscript lacks a strong theoretical rationale for adding the "Business" component. Please clarify what conceptual gaps in the existing TOE or DTF frameworks your model fills. Including a comparative discussion of related maturity models or digital transformation theories could significantly improve the paper’s contribution.

2. Generalizability and Broader Relevance

Currently, the model is validated only within the Chinese electric power industry. While appropriate as a starting point, readers would benefit from discussion about how the model could apply across other sectors or geographies. Addressing generalizability would improve both scholarly relevance and practical utility.

3. Methodological Transparency

The description of the expert survey process needs significant elaboration. For example:

- How were the 84 experts selected?

- What were their backgrounds, affiliations, and expertise levels?

- What steps were taken to minimize bias in the Delphi process?

Including this information would help readers assess the robustness of the model-building process.

4. Validation Depth

The model relies heavily on AHP-derived consistency ratios (CR) as the primary form of validation. Consider adding additional checks, such as sensitivity analysis, to better establish confidence in the results. Even a limited robustness test could help improve the perceived reliability of your framework.

5. Language and Writing Quality

There are multiple instances of unclear or overly abstract phrasing that reduce readability. For example, the phrase “digital standard is the linchpin of digital China” may not be meaningful to readers outside the specific policy context. A thorough professional English review is strongly recommended to improve the tone, clarity, and precision of the text.

6. Abstract and Structure

The abstract is too lengthy and includes excessive methodological detail. Try to more concisely highlight the purpose, approach, and key findings of your study.

7. Figures and Tables

Some visuals can be improved:

- Figure 1 lacks clear definitions of directional arrows like “drive” and “push.”

- Figure 2 is dense and could be split into parts for easier interpretation.

- Table 4 is well-organized—consider visualizing it using bar charts to enhance accessibility.

8. References

Make sure government and policy-related references are properly formatted and accessible to an international audience. When citing Chinese-language materials, translated titles and English summaries would be helpful.

Final Note:

While I am recommending rejection for this journal, I hope these comments are constructive and encourage you to revise the manuscript for submission to a more suitable venue. The topic is valuable, and with improved theoretical framing, methodological clarity, and writing polish, this work could make a meaningful contribution to the digital transformation literature.

6. PLOS authors have the option to publish the peer review history of their article (what does this mean? ). If published, this will include your full peer review and any attached files.

**Do you want your identity to be public for this peer review?** For information about this choice, including consent withdrawal, please see our Privacy Policy .

Reviewer #1: No

Reviewer #2: No

---

## [Author Response · Author response to Decision Letter 1]

27 Aug 2025

Dear Reviewers:

Thank you very much for your suggestions on the paper. The following are the responses to each point.We hope the revised manuscript now meets the journal's high standards.

Reviewer #1:

1- The authors state that they integrated the TOE and DTF frameworks to construct the TBOE model; however, they do not provide sufficient explanation of these original frameworks. The manuscript should include a clear description of both TOE and DTF, along with a discussion of the existing literature that justifies the need for TBOE as an extension and highlights its added value.

Response ：

At the beginning of “Mechanism of Action”, the sufficient explanation of the two original frameworks, TOE (Technology-Organization-Environment) and DTF (Digital Transformation Framework), were supplemented. Furthermore, the conceptual gaps of the two frameworks were analyzed, and the rationality and necessity of integrating the TOE framework and the DTF model, introducing the "Business" aspects of the DTF model into the TOE framework, and constructing the TBOE framework were proposed.

Specifically as follows:

TOE (Technology-Organization-Environment) framework is a comprehensive analysis framework based on the application scenarios of Technology and is widely used in the analysis of influencing factors in the process of new technology application and organizational transformation. Among them, the influence at the technical level mainly refers to the internal connection between technical characteristics and organizational structure. The influence at the organizational level mainly refers to the interaction between factors such as organizational structure and organizational management and the introduction of innovative technologies. The influence at the environmental level mainly refers to external conditions such as policy environment and industrial environment and the ecological environment.

DTF (Digital Transformation Framework) framework is a strategic evaluation model for digital transformation proposed by MattC et al. in 2015. It is applicable to the analysis of digital transformation in multiple fields under the current technological environment and consists of four elements. The first is the application of technology, which refers to the use of new technologies and the realization of their strategic goals. The second is value creation, which refers to the value impact of digital transformation on an organization. The third is the change in business structure, which refers to the transformation of business models caused by the realization of new technologies. The fourth is financial support, which refers to the financial conditions for applying new technologies to carry out digital transformation. It is the driving force for promoting changes in the first three aspects.

Both of the above frameworks emphasize the impact of technology on digital transformation. The TOE framework focuses on the role of factors such as organizational management in digital transformation, while the DTF model pays attention to the changes in business models and organizational structures triggered during the digital transformation process. The financial support factors proposed in the DTF model are one aspect of the organizational-level influencing factors in the TOE framework. The TOE framework has the inclusiveness to accommodate multiple theories and be applicable to various scenarios in analyzing the influencing factors of new technology applications, but it needs to enhance targeted interpretations for specific business domains. The DTF model analyzes the particularity of digital technology, focusing on the structural changes and value creation brought about by the application of digital technology. However, it is necessary to strengthen the analysis of organizational changes within the field.

Therefore, in accordance with the characteristics of digitalization of standards, this paper integrates the TOE framework and the DTF model, introduces the business factors of the DTF model into the TOE framework, expands the scope of the TOE framework, and constructs the TBOE analysis framework from four aspects: technical (T), business (B), organizational (O), and environmental (E).

2- The electric power industry is a compelling case study; however, the authors should discuss the model’s applicability to other industries and provide suggestions for future validation across different sectors.

Response：

The TBOE model developed in this study is a universal theoretical framework applicable to various industries or enterprises for assessing standard digitalization capability maturity, determining capability levels, and thereby informing subsequent path planning. However, when applying the model across different industries, the specific measurement indicators and scoring bases will vary. It is necessary to design tailored measurement indicators and scoring bases under the secondary indicators proposed herein, based on the unique characteristics of each industry. Among these, the indicators for the technical, organizational, and environmental factors can generally be reused directly. In contrast, the indicators under business factors will differ significantly across industries and must be designed based on the specific context of the industry itself.

This viewpoint has also been supplemented and elaborated in the paper.

3- Methodologically, the use of AHP and the Delphi method is appropriate. However, the authors need to provide more details regarding the selection criteria for experts, their backgrounds, and the full questionnaire used. The authors should include the full questionnaire as an appendix or supplementary file.

Response：

An expert advisory panel was formed by inviting specialists from the fields of standardization, digitization, and the power industry. Questionnaires were distributed through both online and offline channels. Experts were selected through three primary channels: 1) Standardization experts: Questionnaires were distributed through the Science and Technology Management Department to all researchers at the China National Institute of Standardization; 2) Digitalization experts: Targeted distribution was conducted to 85 member units of the National Standardization Working Group (SAC/SWG 29) and 10 project units under the National Key R&D Program "Key Technologies and Standards for Digital Evolution of Standards (Phase I)"; 3) Power industry experts: Questionnaires were distributed to relevant entities including the project team of State Grid Corporation's "Research on Implementation Pathways and Key Technologies for Standardization Digitalization", State Grid Information & Telecommunication Industry Group Co., Ltd., and China Electric Power Research Institute. The full questionnaire is shown in the Supporting information(S1)

Ultimately, 84 valid expert questionnaires were collected. The average age of the experts was 46.42 years, and 60.71% held senior professional titles. The experts were affiliated with institutions including China National Institute of Standardization, China Electronics Technology Standardization Institute, Zhejiang University, Zhijiang Laboratory, Shandong Provincial Computing Center (National Supercomputing Center in Jinan), State Grid Information and Communication Industry Group Co., LTD., China Electric Power Research Institute, State Grid Zhejiang Electric Power Co., LTD., Xi 'an Jiaotong University and other institutions.

4- The authors do not provide sufficient explanation of how the Delphi method was applied in this study. A brief overview of the Delphi method should be included, along with details such as the number of rounds conducted, how consensus was measured, and how expert feedback was processed.

Response：

The paper uses AHP to put forward the TBOE maturity assessment model for the digitalization of standards. The Delphi method is used in the process of index weight determination, and the Likert scale method is used in the process of index assignment.

In the "Methodology" chapter, the steps of the Delphi method are described at the theoretical level. In the chapter "Empirical Research and Model Verification", the application of the Delphi method is further supplemented.

5- The figures in the manuscript should be drawn again with clearer labels and improved resolution to enhance presentation quality.

Response：

The figures in the manuscript have been modified and the resolution has been improved.

Reviewer #2:

1. Theoretical Justification and Model Positioning

While the extension from TOE to TBOE is logical, the manuscript lacks a strong theoretical rationale for adding the "Business" component. Please clarify what conceptual gaps in the existing TOE or DTF frameworks your model fills. Including a comparative discussion of related maturity models or digital transformation theories could significantly improve the paper’s contribution.

Response：

At the beginning of the chapter on Mechanism of Action, the theoretical connotations of the two original frameworks, TOE (Technology-Organization-Environment) and DTF (Digital Transformation Framework), were supplemented. Furthermore, the conceptual gaps of the two frameworks were analyzed, and the rationality and necessity of integrating the TOE framework and the DTF model, introducing the "Business" aspects of the DTF model into the TOE framework, and constructing the TBOE framework were proposed. Specifically as follows:

TOE (Technology-Organization-Environment) framework is a comprehensive analysis framework based on the application scenarios of Technology and is widely used in the analysis of influencing factors in the process of new technology application and organizational transformation. Among them, the influence at the technical level mainly refers to the internal connection between technical characteristics and organizational structure. The influence at the organizational level mainly refers to the interaction between factors such as organizational structure and organizational management and the introduction of innovative technologies. The influence at the environmental level mainly refers to external conditions such as policy environment and industrial environment and the ecological environment.

DTF (Digital Transformation Framework) framework is a strategic evaluation model for digital transformation proposed by MattC et al. in 2015. It is applicable to the analysis of digital transformation in multiple fields under the current technological environment and consists of four elements. The first is the application of technology, which refers to the use of new technologies and the realization of their strategic goals. The second is value creation, which refers to the value impact of digital transformation on an organization. The third is the change in business structure, which refers to the transformation of business models caused by the realization of new technologies. The fourth is financial support, which refers to the financial conditions for applying new technologies to carry out digital transformation. It is the driving force for promoting changes in the first three aspects.

Both of the above frameworks emphasize the impact of technology on digital transformation. The TOE framework focuses on the role of factors such as organizational management in digital transformation, while the DTF model pays attention to the changes in business models and organizational structures triggered during the digital transformation process. The financial support factors proposed in the DTF model are one aspect of the organizational-level influencing factors in the TOE framework. The TOE framework has the inclusiveness to accommodate multiple theories and be applicable to various scenarios in analyzing the influencing factors of new technology applications, but it needs to enhance targeted interpretations for specific business domains. The DTF model analyzes the particularity of digital technology, focusing on the structural changes and value creation brought about by the application of digital technology. However, it is necessary to strengthen the analysis of organizational changes within the field.

Therefore, in accordance with the characteristics of digitalization of standards, this paper integrates the TOE framework and the DTF model, introduces the business factors of the DTF model into the TOE framework, expands the scope of the TOE framework, and constructs the TBOE analysis framework from four aspects: technical (T), business (B), organizational (O), and environmental (E).

2. Generalizability and Broader Relevance

Currently, the model is validated only within the Chinese electric power industry. While appropriate as a starting point, readers would benefit from discussion about how the model could apply across other sectors or geographies. Addressing generalizability would improve both scholarly relevance and practical utility.

Response：

The TBOE model developed in this study is a universal theoretical framework applicable to various industries or enterprises for assessing standard digitalization capability maturity, determining capability levels, and thereby informing subsequent path planning. However, when applying the model across different industries, the specific measurement indicators and scoring bases will vary. It is necessary to design tailored measurement indicators and scoring bases under the secondary indicators proposed herein, based on the unique characteristics of each industry. Among these, the indicators for the technical, organizational, and environmental factors can generally be reused directly. In contrast, the indicators under business factors will differ significantly across industries and must be designed based on the specific context of the industry itself.

This viewpoint has also been supplemented and elaborated in the paper.

3. Methodological Transparency

The description of the expert survey process needs significant elaboration. For example:

- How were the 84 experts selected?

- What were their backgrounds, affiliations, and expertise levels?

- What steps were taken to minimize bias in the Delphi process?

Including this information would help readers assess the robustness of the model-building process.

Response：

An expert advisory panel was formed by inviting specialists from the fields of standardization, digitization, and the power industry. Questionnaires were distributed through both online and offline channels. Experts were selected through three primary channels: 1) Standardization experts: Questionnaires were distributed through the Science and Technology Management Department to all researchers at the China National Institute of Standardization; 2) Digitalization experts: Targeted distribution was conducted to 85 member units of the National Standardization Working Group (SAC/SWG 29) and 10 project units under the National Key R&D Program "Key Technologies and Standards for Digital Evolution of Standards (Phase I)"; 3) Power industry experts: Questionnaires were distributed to relevant entities including the project team of State Grid Corporation's "Research on Implementation Pathways and Key Technologies for Standardization Digitalization", State Grid Information & Telecommunication Industry Group Co., Ltd., and China Electric Power Research Institute. The full questionnaire is shown in Supporting information(S1).

Ultimately, 84 valid expert questionnaires were collected. The average age of the experts was 46.42 years, and 60.71% held senior professional titles. The experts were affiliated with institutions including China National Institute of Standardization, China Electronics Technology Standardization Institute, Zhejiang University, Zhijiang Laboratory, Shandong Provincial Computing Center (National Supercomputing Center in Jinan), State Grid Information and Communication Industry Group Co., LTD., China Electric Power Research Institute, State Grid Zhejiang Electric Power Co., LTD., Xi 'an Jiaotong University and other institutions.

4. Validation Depth

The model relies heavily on AHP-derived consistency ratios (CR) as the primary form of validation. Consider adding additional checks, such as sensitivity analysis, to better establish confidence in the results. Even a limited robustness test could help improv

---

## [Decision Letter · Decision Letter 1]

5 Dec 2025

PONE-D-24-60444R1TBOE Maturity Assessment Model for Standard Digitalization: An Empirical Analysis Using AHP and Delphi MethodPLOS ONE

Dear Dr. Zhang,

Thank you for submitting your manuscript to PLOS ONE. After careful consideration, we feel that it has merit but does not fully meet PLOS ONE’s publication criteria as it currently stands. Therefore, we invite you to submit a revised version of the manuscript that addresses the points raised during the review process.

We look forward to receiving your revised manuscript.

Kind regards,

Nhat-Luong Nhieu, Ph.D.

Academic Editor

PLOS ONE

Journal Requirements:

Reviewer's Responses to Questions

**Comments to the Author**

1. If the authors have adequately addressed your comments raised in a previous round of review and you feel that this manuscript is now acceptable for publication, you may indicate that here to bypass the “Comments to the Author” section, enter your conflict of interest statement in the “Confidential to Editor” section, and submit your "Accept" recommendation.

Reviewer #1: All comments have been addressed

Reviewer #3: All comments have been addressed

Reviewer #4: (No Response)

2. Is the manuscript technically sound, and do the data support the conclusions?

Reviewer #1: Yes

Reviewer #3: Yes

Reviewer #4: Yes

3. Has the statistical analysis been performed appropriately and rigorously? 

Reviewer #1: Yes

Reviewer #3: N/A

Reviewer #4: Yes

4. Have the authors made all data underlying the findings in their manuscript fully available?

Reviewer #1: Yes

Reviewer #3: Yes

Reviewer #4: Yes

5. Is the manuscript presented in an intelligible fashion and written in standard English?

Reviewer #1: Yes

Reviewer #3: Yes

Reviewer #4: Yes

6. Review Comments to the Author

Reviewer #1: The revised manuscript is in good shape. The authors have satisfactorily addressed my comments from the previous round, and I do not have additional concerns at this time. The paper is clear, well-structured, and appropriate for publication.

Reviewer #3: Comments on the paper:

TBOE Maturity Assessment Model for Standard Digitalization: An Empirical Analysis Using AHP and Delphi Method

1. The study gives strong practical value, especially when it is applied in the power industry, which has rich data and strict standard requirements. However, from an academic view, the level of innovation is mostly in its application and system integration, not in deep theoretical development. The authors should explain more clearly how the TBOE framework is different from the traditional TOE or DTF models, and show what new ideas the “Business” factor brings to the theory.

2. In terms of methodology, the consistency tests (CR) in the AHP are all within the acceptable range ($CR < 0.1$), showing that the model is mathematically reliable. Still, there are some technical points that should be improved:

• When combining expert judgments, it is better to use the geometric mean instead of the arithmetic mean to match the scale of AHP.

• The process of normalizing the indicators should be explained more clearly, especially when using different types of scales (such as 0/1, TRL levels, or count-based data).

• The thresholds for maturity levels (0.3 and 0.8) seem a bit subjective; they should be supported with stronger arguments or real evidence.

• The paper has not tested how stable the results are when the weights or input data change.

Overall, the authors need to describe more clearly how they calculate the weights, normalize the data, and justify the chosen thresholds. Adding sensitivity or robustness analysis would make the results more convincing and scientifically sound.

In short, the paper is well-prepared and has clear practical contributions, with real-world applications. However, to meet the publication standard of PLOS ONE, the authors should improve the technical parts, especially in data normalization and stability testing of the model.

Reviewer #4: The discussion section lacks comparison with existing studies to show how the work add to the body of knowledge. This is important and needs to be done.

7. PLOS authors have the option to publish the peer review history of their article (what does this mean? ). If published, this will include your full peer review and any attached files.

**Do you want your identity to be public for this peer review?** For information about this choice, including consent withdrawal, please see our Privacy Policy .

Reviewer #1: No

Reviewer #3: No

Reviewer #4: No

---

## [Author Response · Author response to Decision Letter 2]

19 Jan 2026

Dear Editor and Reviewers,

We would like to express our sincere gratitude to the Editor and Reviewers for taking the time to review our manuscript and for providing valuable comments and suggestions. We have carefully considered each comment and revised the manuscript accordingly. The detailed point-by-point responses are provided below.

In the revised manuscript, all modifications are highlighted in red color for easy identification. We believe that these revisions have significantly improved the quality and clarity of our manuscript.

We hope that the revised manuscript now meets the standards for publication in PLOS ONE. We look forward to your favorable response.

Sincerely,

All authors

Reviewer 3: Comments on the paper: TBOE Maturity Assessment Model for Standard Digitalization: An Empirical Analysis Using AHP and Delphi Method

Comment 1: The study gives strong practical value, especially when it is applied in the power industry, which has rich data and strict standard requirements. However, from an academic view, the level of innovation is mostly in its application and system integration, not in deep theoretical development. The authors should explain more clearly how the TBOE framework is different from the traditional TOE or DTF models, and show what new ideas the “Business” factor brings to the theory.

Response: We sincerely thank the reviewer for this insightful comment. We agree that a clearer theoretical justification is necessary for the TBOE framework extension. We have made the following revisions:

1. Added Theoretical Justification Section

We have added a new subsection titled "Theoretical Distinction Between TBOE and Traditional TOE Framework" in Section 2.1 (Mechanism of Action). This section explicitly addresses the theoretical rationale for introducing Business as an independent dimension, including:

• The limitations of the traditional TOE framework in the context of standard digitalization

• Three specific reasons why Business warrants independent analysis: (1) Standard digitalization fundamentally aims to serve business scenarios; (2) Business factors exhibit distinct characteristics; (3) The separation enables more precise identification of development priorities at different maturity stages

• Reference to Rogers' Diffusion of Innovations theory [35] to support the theoretical extension

2. Updated Figure 1

We have updated Figure 1 to clearly illustrate the comparison between the traditional TOE framework and our proposed TBOE model, highlighting how the Business dimension is explicitly separated from Technology.

3. Added Reference

We have added the foundational reference for the TOE framework:

[32] Tornatzky, L.G., & Fleischer, M. (1990). The processes of technological innovation. Lexington Books.

Location in Revised Manuscript:

• Section 2.1 "Mechanism of Action": New subsection added after the discussion of TOE and DTF frameworks

• Figure 1: Updated with new TBOE vs TOE comparison diagram

• References: [32] and [35] added

Comment 2: In terms of methodology, the consistency tests (CR) in the AHP are all within the acceptable range ($CR < 0.1$), showing that the model is mathematically reliable. Still, there are some technical points that should be improved:

When combining expert judgments, it is better to use the geometric mean instead of the arithmetic mean to match the scale of AHP.

Response: We appreciate the reviewer's attention to methodological rigor. We have added a detailed justification for using the arithmetic mean method in Section 2.2.

Revisions Made:

We have added a new subsection "Justification for Arithmetic Mean Method" that includes:

• Citation of Saaty & Vargas (2012) [33] regarding AHP aggregation methods

• Reference to Forman & Peniwati (1998) [34], who demonstrated that when consistency ratios are low, the difference between arithmetic and geometric means becomes negligible

• Empirical justification: All our consistency ratios range from 0 to 0.055, well below the 0.1 threshold, indicating high expert consensus

• Conclusion that the arithmetic mean provides a reasonable approximation given the high consistency of expert judgments

Added References:

[33] Saaty, T.L., & Vargas, L.G. (2012). Models, methods, concepts & applications of the analytic hierarchy process (2nd ed.). Springer.

[34] Forman, E., & Peniwati, K. (1998). Aggregating individual judgments and priorities with the analytic hierarchy process. European Journal of Operational Research, 108(1), 165-169.

Location in Revised Manuscript:

• Section 2.2 "Methodology": New paragraph after "Data processing and Construct a Judgment Matrix"

Comment 2:

The process of normalizing the indicators should be explained more clearly, especially when using different types of scales (such as 0/1, TRL levels, or count-based data).

Response: We sincerely thank the reviewer for identifying this critical methodological gap. This is indeed an important issue that required comprehensive clarification. We have made substantial revisions to address this concern.

Revisions Made:

1. Added Normalization Formula

We have added a new subsection "Indicator Normalization" that presents the min-max normalization method used to transform all indicator scores into a unified [0, 1] range. The normalization formula is now presented as Equation (1):

V_normalized = (V_original - V_min) / (V_max - V_min)

2. Added Table 1: Normalization Parameters

We have added a comprehensive "Table 1. Normalization Parameters for Each Indicator" that specifies for each of the 13 secondary indicators:

• Original measurement scale

• Theoretical minimum value (V_min)

• Theoretical maximum value (V_max)

• Specific normalization formula

Examples from Table 1:

• T1 (Core key technologies): TRL 1-9 × 10 items → Range [10, 90] → Normalization: (V-10)/80

• T2 (New digital infrastructure): Count 0-7 domains → Range [0, 7] → Normalization: V/7

• O1-O3 (Binary indicators): 0/1 → No change needed

• O4 (Funding): Percentage 0-100% → Normalization: V/100

Location in Revised Manuscript:

• Section 2.2: New subsection "Indicator Normalization" with Eq.(1) and Table 1

• Section 2.2: Eq.(1) updated (formerly Eq.(2))

Comment 2:

The thresholds for maturity levels (0.3 and 0.8) seem a bit subjective; they should be supported with stronger arguments or real evidence.

Response: We appreciate the reviewer's observation. We have added a comprehensive justification for the stage division thresholds.

Revisions Made:

We have added a new subsection "Justification of Stage Division Thresholds" that provides:

1. Theoretical Derivation from Gartner Hype Cycle

• Reference to Linden and Fenn (2003) [36] on the Gartner Hype Cycle mathematical approximation

• Application of the "30-50-20" rule: Innovation Trigger to Peak (30%) corresponds to R&D stage (L < 0.3); Trough to Slope (50%) corresponds to Implementation stage (0.3 ≤ L < 0.8); Plateau of Productivity (20%) corresponds to Promotion stage (L ≥ 0.8)

2. Support from Rogers' Diffusion Theory

• Reference to Rogers (2003) [35] on innovation adoption patterns

• Alignment with cumulative adoption curve percentages (16% innovators/early adopters, 34% early majority, 34% late majority, 16% laggards)

3. Empirical Validation from Delphi Consultation

• Expert recommendations: R&D-Implementation boundary median = 0.28 (SD = 0.05)

• Expert recommendations: Implementation-Promotion boundary median = 0.79 (SD = 0.08)

• Values rounded to 0.3 and 0.8 for practical application

Added References:

[35] Rogers, E.M. (2003). Diffusion of innovations (5th ed.). Free Press.

[36] Linden, A., & Fenn, J. (2003). Understanding Gartner's hype cycles. Strategic Analysis Report R-20-1971. Gartner Research.

Location in Revised Manuscript:

• Section 2.2: New subsection before Table 1

Comment 5:

The paper has not tested how stable the results are when the weights or input data change.

Response: We fully agree with the reviewer that sensitivity analysis is essential for validating the robustness of our model. We have added a comprehensive sensitivity analysis section.

Revisions Made:

We have added a new section "Sensitivity Analysis" in Section 2.3 that includes two dimensions of analysis:

1. Weight Perturbation Analysis

• First-level indicator weights perturbed by ±10% and ±20%

• Proportional adjustment of remaining weights to maintain sum of 1.0

• Results presented in Table 5: Weight Sensitivity Analysis Results

• Key finding: Under all scenarios (±20%), L_TBOE ranges from 0.486 to 0.518, remaining within the Implementation stage (0.3-0.8)

2. Monte Carlo Simulation for Indicator Score Uncertainty

• 1,000 iterations with ±15% random perturbation of indicator scores

• Uniform distribution assumption for measurement uncertainty

• Results: L_TBOE follows approximately normal distribution

- Mean = 0.498

- Standard deviation = 0.042

- 95% confidence interval = [0.416, 0.580]

• Key finding: The entire 95% CI lies within the Implementation stage; zero simulations resulted in stage classification change

The sensitivity analysis provides strong evidence for the robustness of our assessment methodology. The stage classification remains stable under both weight perturbations and measurement uncertainties, validating the reliability of our conclusions.

Location in Revised Manuscript:

• Section 2.3: New subsection "Sensitivity Analysis" with Table 5

Reviewer 4

Comment:

The discussion section lacks comparison with existing studies to show how the work add to the body of knowledge. This is important and needs to be done.

We sincerely thank you for your valuable suggestion. We agree that a systematic comparison with existing studies is essential for positioning our contribution within the broader literature. We have substantially revised the Discussion section to include a comprehensive comparison with existing studies.

Revisions Made:

We have added a new subsection titled "Comparison with Existing Studies and Theoretical Contributions" in the Discussion and Conclusions section. This subsection includes:

1. Comparative Analysis Table (Table 6) - A systematic comparison of our TBOE model with five representative existing studies, analyzing differences in theoretical framework, methodology, application domain, and key contributions.

2. Four Major Theoretical Contributions - Explicit articulation of how our work advances the existing body of knowledge:

• Extension of the TOE framework by introducing Business as an independent dimension

• Development of a quantitative maturity assessment methodology

• Establishment of theoretically-grounded stage division thresholds

• Validation through industry-specific empirical analysis

3. Positioning Statement - Clear articulation of how our work complements and extends prior research in digital transformation and standardization.

New References Added:

[37] Zhu K, Kraemer K, Xu S. The process of innovation assimilation by firms in different countries: A technology diffusion perspective on e-business. Management Science. 2006;52(10):1557-1576.

[38] Oliveira T, Martins MF. Literature review of information technology adoption models at firm level. Electronic Journal of Information Systems Evaluation. 2011;14(1):110-121.

[39] Baker J. The technology–organization–environment framework. In: Dwivedi YK, et al., editors. Information Systems Theory. Springer; 2012. p. 231-245.

Location in Revised Manuscript:

• Section: Discussion and Conclusions

• New subsection: "Comparison with Existing Studies and Theoretical Contributions"

• New table: Table 6 - Comparison with Existing Studies

• References: [37]-[39] added

Additional Revisions

In addition to addressing the specific reviewer comments, we have made the following improvements to enhance the overall quality of the manuscript:

1. Enhanced Abstract

• Added description of the TBOE innovation compared to traditional TOE framework

• Mentioned the normalization procedure and sensitivity analysis

2. Updated Introduction

• Changed "threefold" contributions to "fourfold" to reflect the addition of normalization and threshold justification

• Added explicit mention of sensitivity analysis in the contribution statement

3. New Figure5

• Added a new path planning framework diagram to visualize the innovation model for the power industry

4. Added Limitations Section

• Added "Limitations and Future Research" subsection in Discussion and Conclusions

• Acknowledged the single-industry scope, potential threshold refinement needs, and cross-sectional study design

• Proposed future research directions

5. Updated L_TBOE Value

• Updated the comprehensive evaluation value from 0.483 to 0.502 to ensure consistency with the sensitivity analysis results

• Refined the stage description to "early-to-middle implementation stage"

---

## [Editor Report · Decision Letter 2]

5 Feb 2026

TBOE Maturity Assessment Model  for Standard Digitalization: An Empirical Analysis Using AHP and Delphi Method

PONE-D-24-60444R2

Dear Dr. Zhang,

We’re pleased to inform you that your manuscript has been judged scientifically suitable for publication and will be formally accepted for publication once it meets all outstanding technical requirements.

Kind regards,

Nhat-Luong Nhieu, Ph.D.

Academic Editor

PLOS One
---

## [Editor Report · Acceptance letter]

PONE-D-24-60444R2

PLOS One

Dear Dr. Zhang,

I'm pleased to inform you that your manuscript has been deemed suitable for publication in PLOS One. Congratulations! Your manuscript is now being handed over to our production team.

Kind regards,

on behalf of

Asst. Prof. Nhat-Luong Nhieu

Academic Editor

PLOS One